# Identification of *Listeria species* and Multilocus Variable-Number Tandem Repeat Analysis (MLVA) Typing of *Listeria innocua* and *Listeria monocytogenes* Isolates from Cattle Farms and Beef and Beef-Based Products from Retail Outlets in Mpumalanga and North West Provinces, South Africa

**DOI:** 10.3390/pathogens12010147

**Published:** 2023-01-15

**Authors:** Ayanda Manqele, Nomakorinte Gcebe, Rian Ewald Pierneef, Rebone Moerane, Abiodun Adewale Adesiyun

**Affiliations:** 1Department of Production Animal Studies, Faculty of Veterinary Science, University of Pretoria, Onderstepoort, Pretoria 0110, South Africa; 2Agricultural Research Council–Bacteriology and Zoonotic Diseases Diagnostic Laboratory, Onderstepoort Veterinary Research, Private Bag X 05, Onderstepoort, Pretoria 0110, South Africa; 3Agricultural Research Council-Biotechnology Platform, Private Bag X05, Ondersterpoort, Pretoria 0110, South Africa; 4Department of Paraclinical Sciences, School of Veterinary Medicine, The University of West Indies, St Augustine 999183, Trinidad and Tobago

**Keywords:** *Listeria species*, MLVA, *L. monocytogenes*, *L. innocua*, retailers, farms, serogrouping

## Abstract

In this study, *Listeria* isolates (214) were characterized as follows: *L. innocua* (77.10%), *L. monocytogenes* (11.21%), *L. welshimeri* (5.61%), *L. grayi* (1.40%), *L. seeligeri* (0.93%), and L. species (3.73%) that were not identified at the species level, from beef and beef based products from retail and farms in Mpumalanga and North West provinces of South Africa. MLVA was further used to type *Listeria innocua* isolates (165) and *Listeria monocytogenes* isolates (24). The *L. monocytogenes* isolates were also serogrouped using PCR. The MLVA protocol for *L. monocytogenes* typing included six tandem repeat primer sets, and the MLVA protocol for *L. innocua* included the use of three tandem repeats primer sets. The *L. monocytogenes* serogroups were determined as follows: 4b-4d-4e (IVb) (37.50%), 1/2a-3a (IIa) (29.16%), 1/2b-3b (IIb) (12.50%), 1/2c-3c (IIc) (8.33%), and IVb-1 (4.16%). MLVA could cluster isolates belonging to each specie, *L. monocytogenes*, and *L. innocua* isolates, into MLVA-related strains. There were 34 and 10 MLVA types obtained from the MLVA typing of *L. innocua* and *L. monocytogenes*, respectively. MLVA clustered the *L. monocytogenes* isolates irrespective of sample category, serogroups, and geographical origin. Similarly, the *L. innocua* isolates clustered irrespective of meat category and geographical origin. MLVA was able to cluster isolates based on MLVA relatedness. The clustering of isolates from farms and retailers indicates transmission of *Listeria* spp. MLVA is an affordable, simple, and discriminatory method that can be used routinely to type *L. monocytogenes* and *L. innocua* isolates.

## 1. Introduction

*Listeria* spp. are gram-positive, facultative anaerobes, non-spore-forming, flagellated rod-shaped bacterial species that are motile at low temperatures [1,2,3]. *Listeria* species are found to occur ubiquitously in natural environments and can grow in adverse environmental conditions such as unfavorable pH, temperature, and salt content [4,5,6]. The *Listeria* species have been isolated from animals, humans, and food products [7,8]. Since 2020, there have been 21 recognized *Listeria* species and six subspecies [9] (http://www.bacterio.net/listeria.html) accessed 24 November 2022. There are six closely related specie as defined by taxonomy: *Listeria monocytogenes*, *Listeria innocua*, *Listeria ivanovii*, *Listeria welshimeri*, *Listeria seeligeri* and *Listeria grayi* [8,10]. The *Listeria* specie can be divided into two groups based on the species relatedness to *Listeria monocytogenes*, *Listeria sensu lato*, and the *Listeria sensu stricto*; the latter group consists of two pathogenic species (*L. monocytogenes* and *L. ivanovii*) and four non-pathogenic species (*L. innocua*, *L. welshimeri*, *L seeligeri*, and *L. marthii*) [11]. Among the *Listeria* specie, *Listeria monocytogenes* is the pathogenic specie that is responsible for foodborne epidemics and sporadic cases of listeriosis in humans and animals [12]. 

*Listeria monocytogenes* has become a significant foodborne pathogen because of its health implications for humans and the danger it poses to immunocompromised individuals such as pregnant women, the elderly, infants, and immune-suppressed individuals [13]. To those prone to listeriosis infection, the clinical manifestations include encephalitis, septicemia, and miscarriage in pregnant women [14,15]. Even though listeriosis is a rare disease, it has a high case-fatality rate of 20–30%, making *L. monocytogenes* a leading foodborne pathogen associated with human mortality [16]. Unexpectedly, pathogenic *L. monocytogenes* is genetically closely related to *L. innocua*, a non-pathogenic *Listeria* species [17,18]. A whole-genome comparison of *L. monocytogenes* EGDe (serovar 1/2a) and *L. innocua* CLIP 11262 strains by Glaser et al. [19] revealed similarities between the chromosomes of the two strains. *L. innocua* is another widely distributed *Listeria* species found in soil and food products [20]. *Listeria innocua* commonly shares the habitat with *L. monocytogenes* [21]. This makes *L. innocua* eligible to be used as a surrogate for *L. monocytogenes* behavior prediction in food-producing environments [22]. Some studies have suggested the use of *Listeria innocua*, an indicator bacterium for *L. monocytogenes*, in food processing environments [23]. 

*Listeria innocua* has gained interest because of its close relationship with *L. monocytogenes*, as these two are often found co-existing in various environmental, food, and clinical sources [24,25]. These two species are proposed to have evolved from a common ancestor but have come to differ due to the loss of virulence genes in *L. innocua* [26]. It has been further hypothesized that *L. innocua* emerged from the ancestors of the *L. monocytogenes* serogroups four strains and may have possibly retained some of the characteristics of its ancestor [23]. The prfA-virulence gene cluster present in the common ancestor of *Listeria* species was eventually lost in *L. innocua* and *L. welshimeri* [22,23]. The presence of *L. innocua* poses a serious risk as it has been believed to harbor virulence factors transmissible to *L. monocytogenes* [27]. *Listeria innocua* is frequently overrepresented among other Listeria species in food products and processing environments [28]. Establishing the strain types of *L. innocua* circulating in food is therefore important.

Typing pathogens is essential for routine surveillance and the investigation of outbreaks [29]. Therefore, reliable typing methods play a significant role in timely source tracking of pathogens during outbreaks investigations, aiding in the control of the outbreak and halting the spread of infections [30]. There are several methods available for typing *L. monocytogenes*; however, these methods vary in terms of discriminatory power [31]. Conventional serotyping using antisera is the first typing method for *Listeria monocytogenes* serotypes [32]. However, this method is time-consuming and has low discriminatory power [33,34,35]. Molecular typing methods have therefore gained popularity for typing *Listeria* [32]. Various methods have been used for molecular typing of *Listeria*, such as ribotyping, multilocus enzyme electrophoresis (MLEE), pulse field gel electrophoresis (PFGE), and multilocus sequence typing (MLST) [36]. PFGE has been regarded as the ‘gold standard’ method for subtyping *Listeria* due to its discriminatory power, reproducibility, and repeatability [31]. 

Multilocus Variable-number Tandem Repeat Analysis (MLVA) is a Polymerase Chain Reaction (PCR) based typing method that characterizes bacteria through the detection of tandem repeats at several specific loci in the genome of bacteria [37]. This method types bacterial strains by exploiting the fact that the number of repeat units in the variable number of tandem repeats region varies from strain to strain [38]. Among other typing methods, MLVA is considered a rapid, robust, and convenient bacterial subtyping method with reliable discriminatory power for genotyping bacterial isolates [36,39,40]. Since the advent of MLVA for other foodborne pathogens, researchers have used this method for typing bacterial isolates from a variety of food-related sources. Murphy et al. [29] developed an MLVA protocol for typing *L. monocytogenes* isolates from salmon and other related food products. Takahashi et al. [38] developed an MLVA protocol for typing *L. innocua* isolates originating from a food processing facility with the intention of identifying the contamination routes. Miya et al. [41] also used this method to type *L. monocytogenes* serotype 4b isolates that had been previously characterized by PFGE, EcoRI ribotyping, and MLST and concluded that MLVA highly discriminated the isolates. In the current study, we used MLVA to type *L. monocytogenes* and *L. innocua* isolates obtained from beef and beef-based products. Even though typing of *L. monocytogenes* using MLVA has been reported by Gana et al. [42] in South Africa, this is the first report of MLVA typing of *L. innocua* isolates. This study aimed to subtype *L. innocua* and *L. monocytogenes* isolates using MLVA. 

## 2. Materials and Methods

### 2.1. Origin of Isolates

*Listeria* isolates used in this study originated from farms and retail outlets in Mpumalanga and Northwest Provinces, South Africa. A total of 214 samples were collected over one year, from November 2020 to December 2021. Samples collected were fecal and feed samples from farms and beef and beef-based products from retail outlets. The isolation and detection of *Listeria* isolates were performed as described by Matle et al. [43]. Briefly, 25 g of each sample was added into 225 mL of One Broth-*Listeria* (Oxoid, Basingstoke, UK) in a stomacher bag, and the contents were homogenized at high speed for 30 s using a Stomacher (Stomacher Lab Blender 400, Seward Ltd., West Sussex, UK). The mixture was incubated at 37 °C for 24 h. Post incubation, a loop full was inoculated onto Brilliance *Listeria* agar (BLA) (Oxoid, Basingstoke, UK) and incubated at 37 °C for 48 h. The presumptive *Listeria* colonies were preserved in 70% glycerol and stored at −80 °C for further analysis. 

### 2.2. Microbiological Resuscitation of Listeria Isolates 

The glycerol preserve was thawed at room temperature. The mixture was vortexed at high speed for 30 s, and 1 mL of the mixture was inoculated into 9 mL brain heart infusion (BHI) broth (Oxoid, Basingstoke, UK) and incubated at 37 °C for ±24 h. Post-incubation, the enrichment was plated onto Brilliance *Listeria* agar (BLA) (Oxoid, Basingstoke, UK) plates and incubated at 37 °C for 48 h. After that, a single colony was picked and re-subcultured onto Brilliance *Listeria* agar (Oxoid, Basingstoke, UK) and incubated at 37 °C for 48 h. The pure culture was then subjected to DNA extraction using the cell-lysis boiling method.

### 2.3. DNA Extraction Using Cell-Lysis Boiling Method

DNA was extracted using the cell lysis boiling method adopted from Madoroba [44]. A loopful of culture was suspended in 200 µL of distilled water, and the suspension was vortexed at high speed for 30 s. The mixture was boiled at 96 °C for 10 min and allowed to cool for 5 min at room temperature. Post-cooling, it was centrifuged for 5 min at 15,493× *g*, the supernatant containing the DNA was transferred into a clean microcentrifuge tube, and the pellet was discarded. The supernatant, containing DNA, was kept and stored at −20 °C for later use in PCR assays.

### 2.4. Listeria Speciation Using PCR

With minor modifications, *Listeria* speciation was conducted according to the PCR method by Ryu et al. [45]. The multiplex PCR was designed for the speciation of *Listeria* species into *L. grayi*, *L. innocua*, *L. ivanovii*, *L. monocytogenes*, *L. seeligeri*, and *L. welshimeri*. Each 25 µL reaction contained 10 µL 2X Red Taq Mastermix (Ampliqon A/S, Odense, Denmark), 4 µL of 20 mM primer mix, 4 µL DNA template, and 7 µL of PCR water. The following thermocycler conditions were used: denaturation at 94 °C for 5 min, 35 cycles of denaturation at 94 °C for the 30 s, annealing at 60 °C for 30 s, extension at 72 °C for 32 s, and final extension at 72 °C for 5 min. *L. monocytogenes* ATCC 19111 strain, *L. grayi* ATCC 19120 strain, *L innocua* ATCC 11288 strain, *L. ivanovii* ATCC 19119 strain, *L. seeligeri* ATCC 35967 strain, and *L. welshimeri* ATCC 35897 strain were used as a positive control, and *E. coli* strain ATCC 2922 served as a negative control, and water was used as blank. The amplicons were subjected to gel electrophoresis using 1% agarose gel containing ethidium bromide and 100 V for 1 h 45 min. A 100 bp DNA ladder (Thermo Fischer Scientific; Waltham, MA, USA) was used to estimate amplicon sizes. Post electrophoresis, the gel was visualized under ultraviolet light, and the image was captured using a gel documentation system (BIORAD; Hercules, CA, USA).

### 2.5. PCR Serogrouping of L. monocytogenes

A PCR method described by Doumith [12] was used for the serogrouping of *L. monocytogenes* isolates. This method included the use of five primer pairs *OFR2819* (471bp), *OFR2110* (597bp), *lmo0737* (691bp), *lmo1118* (906bp), and *prs* (370bp). The PCR assay was performed as a multiplex PCR in a 25 µL reaction with the following PCR reagents: 12.5 µL 2X Red Taq Mastermix (Ampliqon A/S, Odense, Denmark), 4 µL of 20 mM primer mix, 3.5 µL of PCR water and 5 µL DNA template. The PCR was conducted under the following thermocycler conditions: denaturation at 94 °C for 3 min, 35 cycles of 94 °C for 40 s, annealing at 53 °C for 1.15 min, extension for 72 °C at 1.15 min, and final extension at 72 °C for 7 min. *L. monocytogenes* ATCC 19111 strain was used as a positive control, *E. coli* strain ATCC 25922 was used as the negative control, and water was used as blank. The amplicons were subjected to gel electrophoresis using 3% agarose gel containing ethidium bromide and 100 V for 3 h. A 100 bp DNA ladder (Thermo Fischer Scientific; Waltham, MA, USA) was used to estimate amplicon sizes. The gel was visualized under ultraviolet light, and the image was captured using a gel documentation system (BIORAD; Hercules, CA, USA).

### 2.6. MLVA Typing of L. monocytogenes 

MLVA typing of *L. monocytogenes* was performed as a singleplex PCR method according to a protocol developed by Murphy [29] for typing *L. monocytogenes* isolates. The current study used six primer pairs (TR-1, TR-2, TR-3, TR-4, TR-5, and TR-6) of 20 mM concentration. A 25 µL PCR assay contained 12.5 µL of 2X Red Taq Mastermix (Ampliqon A/S; Odense, Denmark), 2 µL Forward primer, 2 µL Reverse primer, 6.5 µL of PCR water, and 2 µL of template DNA. The thermocycler conditions were 35 cycles of 94 °C for 45 s, an annealing temperature of 54° for primers TR-1 to TR-5 (the annealing temperature for TR 6 primers was only 52 °C), and 72 °C for 60 s, with a final extension at 72 °C for 5 min. The PCR assay included a positive control, *L. monocytogenes* strain (ATCC 19111), *E. coli* ATCC 25922 as a negative control, and water as a blank. The PCR amplicons were resolved in conventional gel electrophoresis using a 3% agarose gel stained with ethidium bromide running at a voltage of 80 for 4 h. A 50 bp ladder (New England, Biolabs) was included for the estimation of the sizes of the amplicons. The gel images were captured using a gel documentation system (BIORAD; Hercules, CA, USA).

#### Determination of MLVA Types

As described by Murphy et al. [29], TR copy numbers were determined. The amplicon fragment size and TR size were used to calculate each Tandem Repeat copy number. Each amplified DNA was then allocated an allele number. Each isolate was given an MLVA profile (MLVA allele string) using the number of tandems repeats at each locus (TR-1, TR-2, TR-3, TR-4, TR-5, and TR-6). Each isolate was assigned under MLVA types using unique MLVA profiles. The UPGMA software was used to plot the minimum spanning trees and construct the dendrograms to elucidate the genetic relatedness of the MLVA types.

### 2.7. MLVA Typing of L. innocua

An MLVA typing method for *L. innocua* developed by Takahashi et al. [38] was applied in the current study. Three primer sets, designated TR-D, TR-E, and TR-J (Table 1), were used for typing by targeting tandem repeats. The PCR amplification was performed in singleplex. The PCR reaction was prepared to a final volume of 25 µL with the following reagents: 12.5 µL 2X Red Taq Mastermix (Ampliqon A/S, Odense, Denmark), 1.25 µL (1 Mm) Forward Primer, 1.25 µL (1 Mm) Reverse Primer, 7 µL water and 3 µL DNA. Amplification was performed under the following thermocycler conditions: 94 °C for 5 min, 30 cycles of 94 °C for 30 s, 60 °C for 40 s, and 72 °C for 1 min. *Listeria innocua* ATCC 33090 strain was used as a positive control, *E. coli* ATCC 25922 as a negative control, and PCR water was used as blank. The PCR products were separated using electrophoresis 3% ethidium bromide-stained agarose gel at 80 Voltage for 4 h. A 50 bp ladder (New England, Biolabs, Ipswich, MA, USA) was included for the estimation of the sizes of the amplicons. The gel was visualized under ultraviolet light, and the image was captured using a gel documentation system (BIORAD; Hercules, CA, USA).

#### Determination of the MLVA Types

Each MLVA type was determined as described by Takahashi et al. [38]. First, the Amplified DNA size was estimated and used to calculate the TR copy number. The allele number for each TR locus was calculated by a formula that included the size of the flanking regions of the 3′ end and the 5′ divided by the size of the tandem repeat. The calculations were applied for all tandem repeats, and an allele string was determined for TR-D, TR-E, and TR-J. The allele string was used to allocate each isolate an MLVA type. The UPGMA software was used for plotting the minimum spanning trees and constructing the dendrograms to elucidate the genetic relatedness of the MLVA types.

### 2.8. Data Analysis

The chi-square test was used to test the significant difference of the MLVA types according to geographical location and sample category. A significant test (*p* < 0.05) indicated significant variation in compared categories. 

## 3. Results

### 3.1. Determination of Listeria Species

A total of 214 isolates from retail outlets and farms were classified into *L. innocua* (77.10%, n = 165), *L. monocytogenes* (11.21%, n = 24), *L. welshimeri* (5.61%, n = 12), *L. grayi* (1.40%, n = 3), *L. seeligeri* (0.93% n = 2), and *Listeria* species which were not identified to species level (3.73%, n = 8) (Table 2). Among the six *Listeria* species targeted, *L. ivanovii* was not detected in all the isolates. Figure 1 illustrates the percentage of detection of each of the *Listeria* specie. Among the sample categories, raw meat was the only category with all the *Listeria* species detected from a variety of its samples. Furthermore, all the *Listeria* isolates not fully characterized into species level were detected from raw meat samples. The other sample categories, ground meat, offals, processed meat, and RTE, all had multiple *Listeria* species detected per category. Fecal and feed sample categories were the only categories that detected *L. innocua* from all the samples found in these categories (Table 2).

### 3.2. Detection of Serogroups of L. monocytogenes Isolates 

The 24 *L. monocytogenes* isolates were grouped into five serogroups 4b-4d-4e (IVb) (37.50%, n = 9), 1/2a-3a (IIa) (29.16%, n = 7), 1/2b-3b (IIb) (12.50%, n = 3), 1/2c-3c (IIc) (8.33%, n = 2), and IVb-1 (4.16%, n = 1). Two (8.33%) isolates could not be serogrouped using the PCR method described by [12].

### 3.3. MLVA Typing of L. monocytogenes

MLVA was able to type and discriminate *L. monocytogenes* (24) isolates from beef and beef-based products from the retail outlets according to MLVA-type relatedness. The isolates did not cluster according to meat type, meat category, serogroups, or province of origin. The clusters showing relatedness of the isolates are illustrated in the dendrogram and minimum spanning tree in Figure 2 and Figure 3. Ten MLVA patterns were identified from *L. monocytogenes* isolates (Table 3). The frequently detected MLVA pattern was MLVA type 1 (20.83% n = 5). This pattern was detected from beef meat (processed meat, offal, raw meat, and ground meat) from both provinces, and the isolates belonged to serogroups IVb and IIa. The other MLVA types that were detected were as follows: MLVA type 2 (4.16%, n = 1), MLVA type 3 (8.33%, n = 2), MLVA type 4 (12.5%, n = 3), MLVA type 5 (16.66%, n = 4), MLVA type 6 (16.66%, n = 4), MLVA type 7 (8.33%, n = 2) MLVA type 8 (4.16%, n = 1), MLVA type 9 (4.16%, n = 1), and MLVA type 10 (4.16%, n = 1). There was no statistical significance association (*p* = 0.409) among the MLVA types that were detected between the two provinces. There was also no statistically significant association (*p* = 0.451) among the MLVA types that were detected among the sample categories. MLVA typing discriminated isolates irrespective of the serogroups. For instance, serogroups IIa had seven isolates that could be grouped into five MLVA types, serogroups IIb had three isolates that could be grouped into three MLVA types, serogroups II c had two isolates that could be characterized into two MLVA types, and serogroups IVb had nine isolates that could be categorized into four MLVA types. Out of the six tandem repeats used, only TR-1 and TR-3 showed variation in the copy number. TR-2, TR-5, and TR-6 had no variation in each of the copy numbers that were obtained, and TR-4 had zero copy numbers for all the isolates. 

### 3.4. MLVA Typing of L. innocua

The MLVA-typing method was able to discriminate the *L. innocua* isolates based on the MLVA-type relatedness. Thirty-four MLVA types were identified from 165 *L. innocua* isolates. The clustering of the isolates was irrespective of meat category, meat type, and geographical location, as shown in the dendrogram and minimum spanning tree in Figure 4 and Figure 5. The predominant MLVA type 15 with an MLVA profile (14-17-11) was detected in 21% (35/165) of the isolates. These isolates were not limited to a certain type of meat but were represented in all meat types and both provinces (Table 4). MLVA also revealed an increased variety of profiles in both farms and retailers. A total of 9 (26.47%) of the MLVA types (3, 4, 12, 14, 15, 17, 23, 29, and 31) originated from isolates from both cattle farms and retailers. A total of 9 (26.47%) MLVA patterns (1, 11, 13, 21, 22, 25, 27, 28, and 30) originated from cattle farms only, and 16 (47.05%) MLVA patterns (2, 5, 6, 7, 8, 9, 10, 16, 18, 19, 20, 24, 26, 32, 33, and 34) were from beef products only from retailers (Table 4). There was a statistically significant association (*p* = 0.002) in the MLVA types that were detected among sample categories. These MLVA patterns also exhibited a genetic relationship among isolates recovered from cattle farms and retail outlets, irrespective of the province of origin. MLVA types 1, 6, 7, 8, 10, 11, 16, 17, 21, 23, 24, 26, 27, 28, 29, 30, and 31 were only detected from isolates from Mpumalanga. MLVA types 2, 5, 9, 18, 20, and 25 were detected from North West only. However, there was no statistically significant association (*p* = 1.835) among MLVA types that were detected between Mpumalanga and North West Provinces. For the three tandem repeat primers (TR-D, TR-E, and TR-J) used, all the tandem repeats showed variation in tandem repeat copy numbers. However, TR-D and TR-E had 20 and 2 isolates, respectively, that had zero copy numbers. The isolates that had zero copy numbers for TR-D were from a variety of retail samples and one fecal sample, while the samples that had zero copy numbers for TR-E were from two beef stewing samples originating from Mpumalanga. 

## 4. Discussion

*Listeria* species are widely distributed in nature and, therefore, effortlessly found in various food products [46]. Multiple *Listeria* species have been detected in beef meat [47]. In the current study, beef-based products were found contaminated with different *Listeria* species, including the pathogenic *L. monocytogenes*, non-pathogenic *L. grayi*, *L. welshimeri*, *L. seeligeri*, and *L. innocua*. Both pathogenic and non-pathogenic strains of *Listeria* can be found inhabiting the same environment [48]. Similarly, Gebremedhin et al. [49] reported the presence of multiple *Listeria* species from raw meat and ready-to-eat (RTE) meat from Ethiopia. The presence of any *Listeria* specie in food products is an indication of poor hygiene [50]. Serogroup IVb of *L. monocytogenes* was overrepresented among serogroups. This serogroup includes serotype 4b, a serotype frequently identified in human listeriosis cases [51]. Isolates belonging to serogroup IVb have been associated with several clinical cases in humans [52]. The serogroups identified from the current study have been detected from various meat products by Matle et al. [43] and equipment and utensils from food processing environments [53]. The presence of these serotypes in food is a severe health risk as they are commonly transmitted through food to humans [54].

The first report on MLVA subtyping of *L. monocytogenes* in South Africa was published by Gana et al. [42]. However, the MLVA patterns found in these authors’ study were not similar to ours. These authors identified 16 MLVA types, slightly higher than the MLVA types reported in our study, possibly due to differences in sample sizes. This also suggests that the *L. monocytogenes* strains circulating in Gauteng are not entirely the same as those found in Mpumalanga and North West based on the MLVA analysis. There is currently no database that can be used as a reference for *L. monocytogenes* MLVA types. In the current study, MLVA was employed for subtyping *L. monocytogenes* from retail outlets. MLVA was able to discriminate the *L. monocytogenes* isolates in the current study, as other authors have used this method for the discrimination of *Listeria monocytogenes* isolates from various sources [55,56]. There were 10 MLVA patterns for *L. monocytogenes*; these results concurred with the results presented by Saleh-Lakha et al. [37], as these authors produced 11 MLVA patterns from a culture collection of 2,019 isolates from various food, environmental and clinical sources from Ontario, Canada. However, the results in the current study are different from the 27 MLVA patterns reported by Martín et al. [31] from food products (raw meat, RTE) and food processing plant contact surfaces in Spain, which might be attributed to the use of eight primer pairs and a larger sample size compared to our study that had a protocol with six primer pairs. 

MLVA displayed a higher discriminatory power compared to PCR serogrouping since each serogroup was characterized from the isolates. There were multiple MLVA types that were detected except for serogroup IVb-1, as there was one isolate belonging to this serogroup. This is indicative of the higher discriminatory power of MLVA compared to serogrouping. It is also worth mentioning that none of the isolates clustered according to meat type, meat category, or geographic origin of the isolates. In our study, MLVA clustered isolates irrespective of their serogroups, sample type, and geographical origin. Similarly, the MLVA study reported by Gana et al. [42] clustered *L. monocytogenes* isolates irrespective of serogroups, sample type, and source. This highlights the ability of MLVA to cluster purely based on MLVA relatedness. There were only two (LM-TR-1 and LM-TR-3) that played a significant role in the discrimination of *L. monocytogenes* isolates by displaying variation in the tandem repeats copy number of the isolates. These played a significant role in the discrimination of the *Listeria* isolates. LM-TR-2, LM-TR-5, and LM-TR-6 did not display any variation as each TR produced a uniform copy number for all the isolates, and lastly, LM-TR-4 had zero copy numbers detected for all the isolates. The results of uniform copy numbers obtained for LM-TR-2 are similar to the copy numbers obtained by Gana et al. [42] for this tandem repeat.

To our understanding, this study reports the first MLVA typing of *Listeria innocua* in South Africa. The first MLVA typing for the *L. innocua* method was developed by Takahashi et al. [38]. These authors further used this protocol to type *L. innocua* isolates from the food processing environment and identify the MLVA types of these isolates. MLVA is an appropriate subtyping tool as this is a rapid, highly discriminatory method that types bacterial isolates based on detecting tandem repeats at several specific loci [37]. *Listeria innocua* is frequently found overgrowing *L. monocytogenes* in food and environmental samples rendering *L. innocua* an important pathogen. The increased number of MLVA types obtained from typing *L. innocua* isolates indicates the high genetic diversity of the *L. innocua* specie from retailers and farms. Furthermore, the variety of the MLVA types detected in farms and retailers is evident that according to MLVA, the strains are diverse even though they belong to *L. innocua* and originate from bovine-related samples. The predominant MLVA type 15 was obtained from isolates originating from both retailers and farms; this could indicate that this type is highly prevalent in farms and retailers among bovine-related samples. This could also mean that this sequence type is freely circulating in the farm environment and may be transmitted from farms to retail outlets. There were more MLVA types from retail outlets compared to farms, with Mpumalanga having more MLVA types variety compared to North West. This may be attributed to the increased number of samples collected from Mpumalanga (n = 117) compared to the North West (n = 48). All the tandem repeats primers (TR-D, TR-E, and TR-J) used in our study provided enough variation to discriminate *L. innocua* isolates. This is similar to Takahashi [38] when the authors used this method to discriminate *L. innocua* isolates from the food-producing environment after optimizing the MLVA for *L. innocua* protocol. We also detected 20 isolates that had zero tandem repeats copy numbers for TR-D and zero-tandem repeat copy number for TR-E. This is different from the results by Takahashi et al. [38], as these authors did not have any isolates that had zero copy numbers for any of the tandem repeats. 

## 5. Conclusions

We can conclude that detecting *Listeria* species, both pathogenic and non-pathogenic, from raw and RTE beef products calls for serious concern as these pose serious health risks to humans since there are rare clinical cases of non-pathogenic species. *L. innocua* is the most overrepresented *Listeria* species compared to other species. Even though this bacterium is non-pathogenic, its presence in food and food environments should be taken seriously as it is frequently found sharing the same environment with *L. monocytogenes*. The MLVA subtyping method applied in the current study has proven to be a reliable subtyping method that is fast and affordable. MLVA was able to distinguish the type of *L. monocytogenes* and *L. innocua* strains based on tandem repeat analysis. The superiority of this method compared to serotyping was also displayed in this study. Therefore, this method can be used in surveillance studies and monitoring studies. MLVA has proven to type *Listeria innocua* and *L. monocytogenes* and cluster the isolates irrespective of the source and serogroups. 

## Figures and Tables

**Figure 1 pathogens-12-00147-f001:**
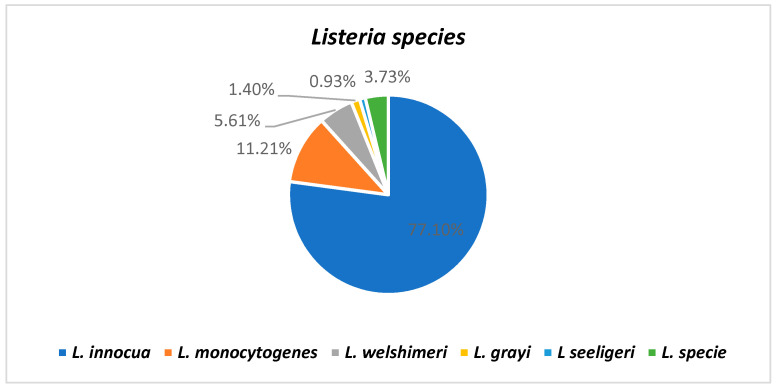
Distribution of *Listeria* species among the samples.

**Figure 2 pathogens-12-00147-f002:**
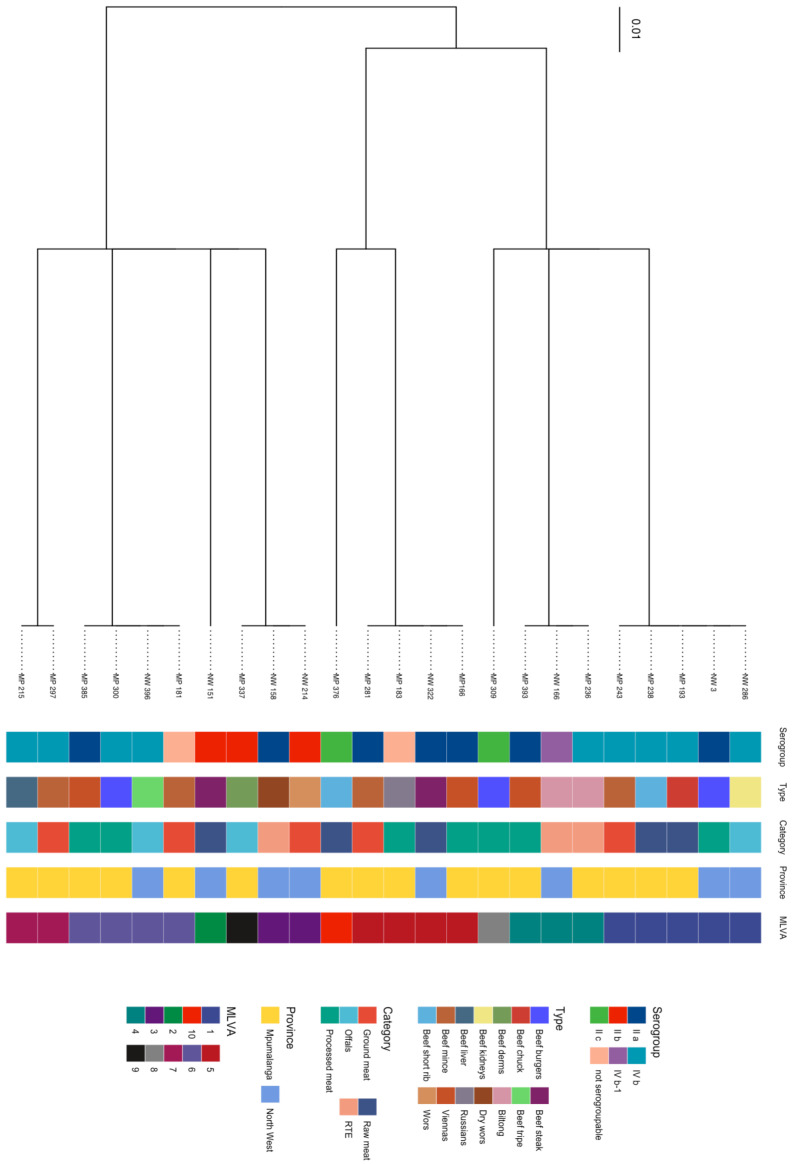
UPGMA Dendrogram of *L. monocytogenes* isolates obtained from retail outlets in Mpumalanga and North West.

**Figure 3 pathogens-12-00147-f003:**
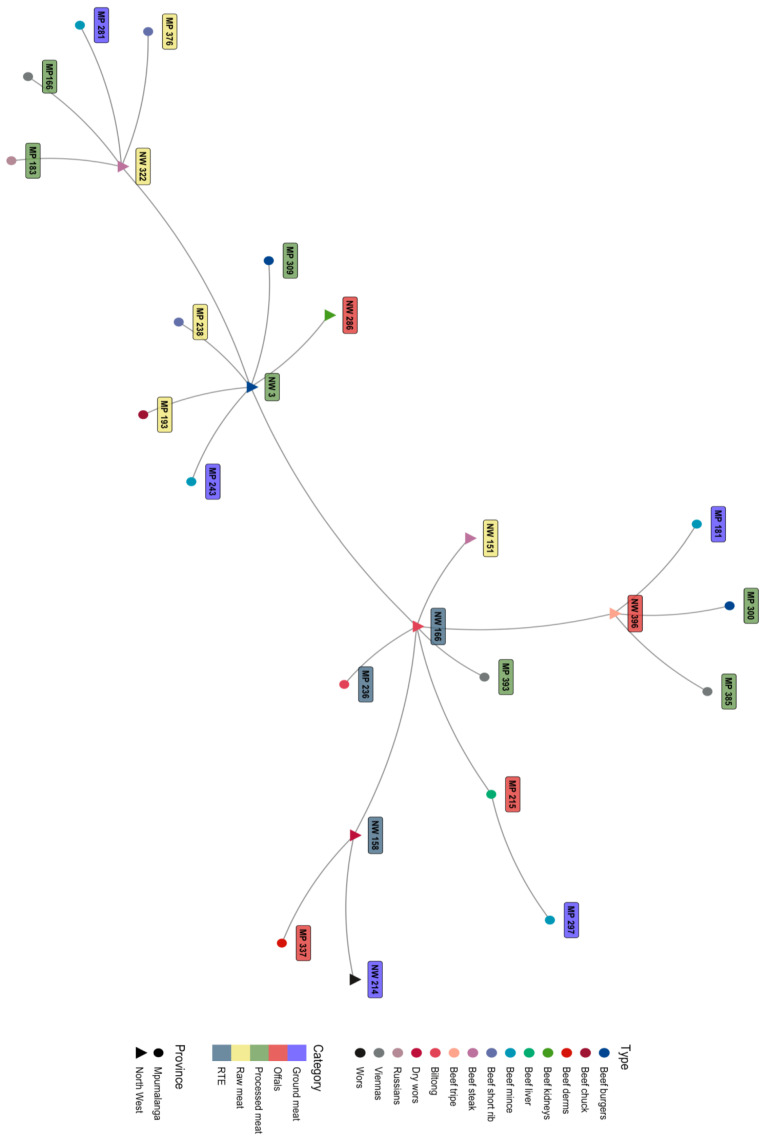
Minimum Spanning Tree displaying the relationship of *L. monocytogenes* isolates obtained from retailers in Mpumalanga and North West.

**Figure 4 pathogens-12-00147-f004:**
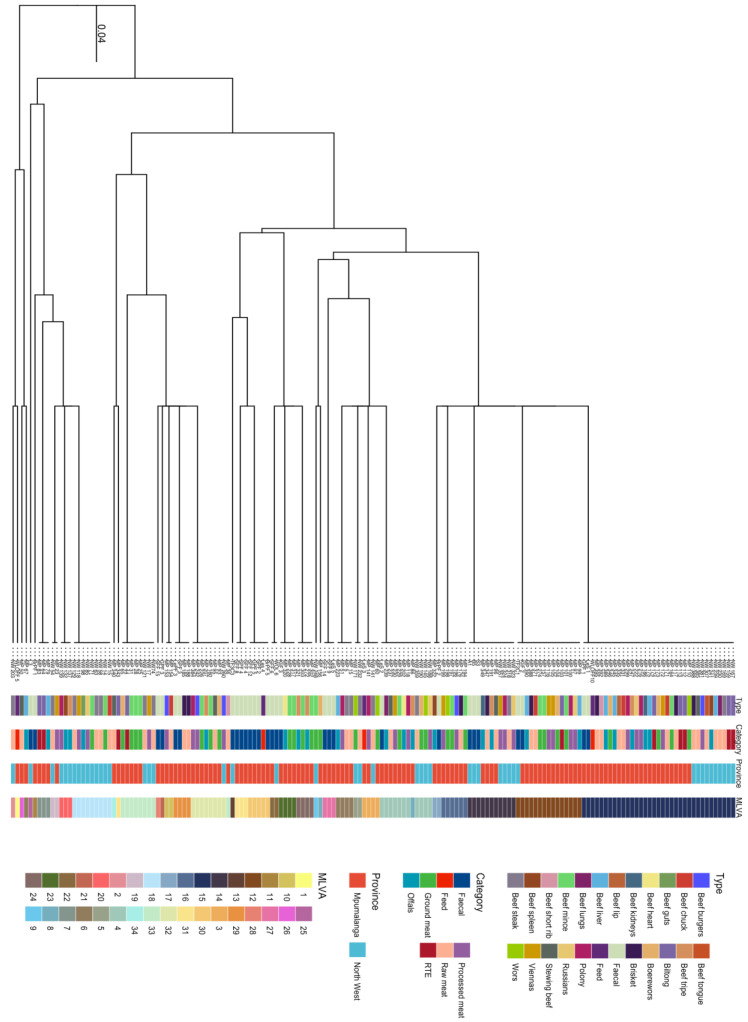
UPGMA Dendrogram of *L. innocua* isolates obtained from farms and retail outlets in Mpumalanga and North West.

**Figure 5 pathogens-12-00147-f005:**
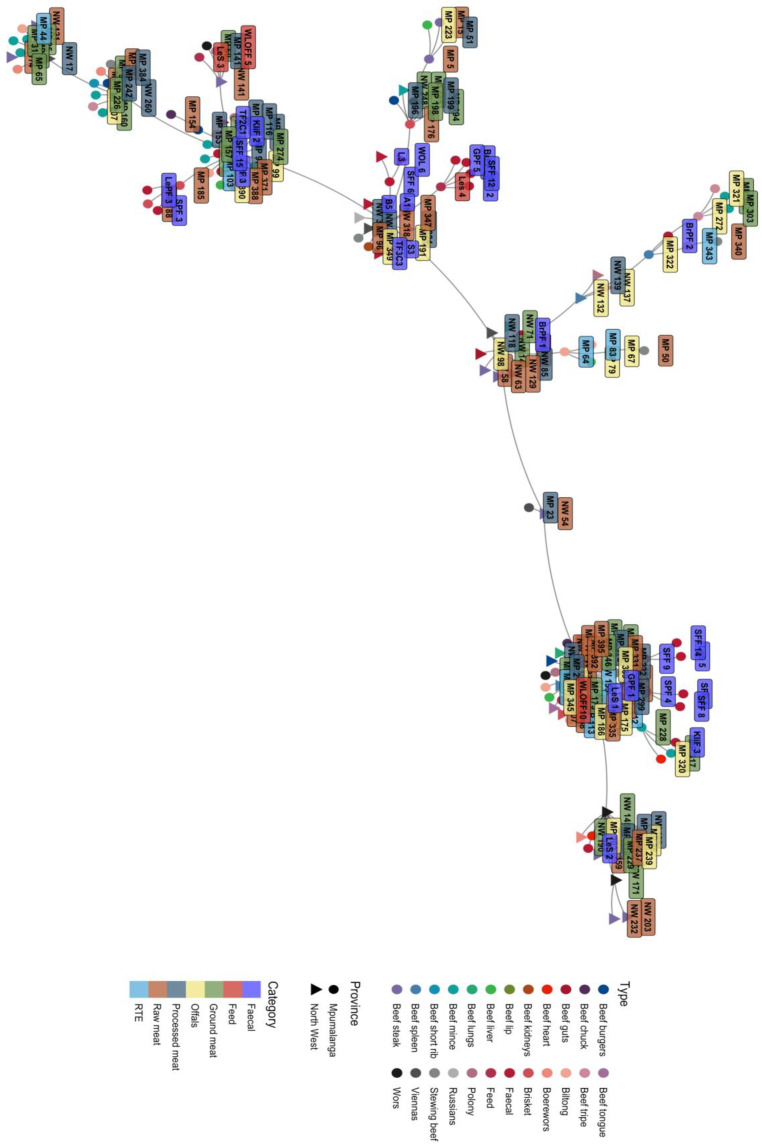
Minimum Spanning Tree, displaying the relationship of *L. innocua* isolates from retailers and farms in Mpumalanga and North West.

**Table 1 pathogens-12-00147-t001:** Primers used for MLVA typing of *L. innocua*.

Tandem Repeats (TR)	Primer	Sequence 5′-3′
TR-D	D-Forward	GACAAAAGTAAGTCATGCGGGTATTT
	D-Reverse	TAGCTACAATCGGATTAACGG
TR-E	E-Forward	GTACCTCCATTTGCTGTTCCA
	E-Reverse	ATGTTATCCACCTTCAAGTAACTG
TR-J	J-Forward	ATGTTTGTGTTCTCAGTTGCC
	J-Reverse	CTACCAAGGATTACTACAAGAAC

**Table 2 pathogens-12-00147-t002:** *Listeria* species identified from different sample categories.

Sample Category	*L. grayi*	*L. innocua*	*L. monocytogenes*	*L. seeligeri*	*L. welshimeri*	*L. specie* *	Total
Fecal		29					29
Feed		4					4
Ground meat	2	31	5		2		40
Offals		24	4	1	4		33
Processed meat		28	7		1		36
Raw meat	1	39	5	1	5	7	58
RTE		10	3			1	14
Total	3	165	24	2	12	8	214

* could not be identified to species level.

**Table 3 pathogens-12-00147-t003:** MLVA patterns of *L. monocytogenes* isolates.

MLVA Type	MLVA Pattern ^a^	Serogroup(s)	Meat Category	Frequency of Occurrence. % (n)	Geographical Origin ^b^
1	04-02-20-00-27-04	IIa, IVb	Processed meat, Offals, Raw meat, Ground meat	20 (5)	NW, MP
2	05-02-14-00-27-04	IIb	Raw meat	4 (1)	NW
3	05-02-06-00-27-04	IIa, IIb	RTE, Ground meat	8 (2)	NW
4	05-02-20-00-27-04	IVb-1, IVb, IIa	RTE, Processed meat	13 (3)	NW, MP
5	04-02-25-00-27-04	IIa, NS	Raw meat, Processed meat, Ground meat	17 (4)	NW, MP
6	05-02-25-00-27-04	IVb, IIa, NS	Offals, Ground meat	17 (4)	NW, MP
7	05-02-09-00-27-04	IVb	Offals, Ground meat	8 (2)	MP
8	11-02-20-00-27-04	IIc	Processed meat	4 (1)	MP
9	04-02-06-00-27-04	IIb	Offals	4 (1)	MP
10	11-02-25-00-27-04	IIc	Raw meat	4 (1)	MP

^a^ MLVA pattern (TR-1, TR-2, TR-3, TR-4, TR-5, and TR-6). ^b^ Geographical origin, NW—North West and MP—Mpumalanga Provinces.

**Table 4 pathogens-12-00147-t004:** MLVA patterns of *L. innocua* isolates.

MLVA Type	MLVA Pattern ^a^	Meat Category/Farm Sample ^c^	Frequency of Occurrence. % (n)	Geographical ^b^ Origin
1	08-09-05	Feed	0.6 (1)	MP
2	08-09-12	RM	0.6 (1)	NW
3	08-17-05	RM, feed, GM, PM,	2.4 (4)	NW, MP
4	08-17-11	GM, PM, offals, RM, feces	6.6 (11)	NW, MP
5	08-17-12	GM, RM	1.2 (2)	NW
6	08-17-16	RM, PM, offals	3.0 (5)	MP
7	08-25-08	RTE, offals	1.8 (3)	MP
8	09-17-11	GM, offals	1.2 (2)	MP
9	09-17-16	GM	0.6 (1)	NW
10	10-17-05	PM, RM	1.2 (2)	MP
11	14-09-08	feces	0.6 (1)	MP
12	14-17-05	PM, offals, GM, RTE, feces	9 (15)	NW, MP
13	00-17-08	feces	0.6 (1)	NW
14	14-17-18	RM, PM, offals, feces	6.6 (11)	NW, MP
15	14-17-11	RM, PM, offals, GM, RTE, feces, feed	21 (35)	NW, MP
16	14-17-16	RM, GM, PM	3.6 (6)	MP
17	14-17-14	Offals, feces	1.2 (2)	MP
18	14-25-08	GM, RM, PM	5.4 (9)	NW
19	14-25-11	RM, PM	1.2 (2)	NW, MP
20	14-25-14	PM, offals	1.8 (3)	NW
21	14-25-14	feces	06. (1)	MP
22	19-17-05	feces	1.2 (2)	NW, MP
23	19-17-11	GM, offals, feces	1.8 (3)	MP
24	19-17-14	GM, Offals	2.4 (4)	MP
25	19-20-08	feces	0.6 (1)	NW
26	08-00-16	RM	0.6 (1)	MP
27	30-17-11	feces	1.8 (3)	MP
28	30-17-05	feces	0.6 (1)	MP
29	36-17-05	RM, feces	2.4 (4)	MP
30	36-17-08	feed, feces	3.0 (5)	MP
31	36-17-11	RTE, feces	2.4 (4)	MP
32	00-17-05	PM, GM, RM, offals	4.8 (8)	NW, MP
33	00-25-05	PM, RM	5.4 (9)	NW, MP
34	00-00-14	RM	0.6 (1)	MP

^a^ MLVA allele string (TR-D, TR-E, TR-J); ^b^ Geographical origin (Mpumalanga (MP), North West Provinces (NW); ^c^ Meat category/farm samples, RM—raw meat, GM—ground meat, PM—processed meat, RTE—Ready-to-eat meat.

## Data Availability

Not applicable.

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
