# Peer review of "Identification of Listeria species and Multilocus Variable-Number Tandem Repeat Analysis (MLVA) Typing of Listeria innocua and Listeria monocytogenes Isolates from Cattle Farms and Beef and Beef-Based Products from Retail Outlets in Mpumalanga and North West Provinces, South Africa"

_pathogens, 2023, doi:10.3390/pathogens12010147_

Round 1
Reviewer 1 Report
The authors presented an identification and subtyping of Listeria monocytogenes and Listeria innocua from beef production and beef products
I believe that MLVA typing does not provide much information about strains, nowadays more advanced and detailed characterization is affordable and provides much more. However, the research that has been conducted was well planned and number of samples well represents beef production and the possibility of Listeria occurrence.
The manuscript can be improve and require some corrections
Title: I believe that it is worth to precise the isolation source
Introduction:
line 42: link doesn't work
line 101: Sentence started from the citation, it should be last name of the first author, this should be done in whoel text
Materials and Methods
line 114: numbers of samples should be provided
line 126: lack of producer of BHI
Results:
line 226: capital letter in illustrate
3.1. determination of Listeria species: Table 2 should be described, there is a lack of information about the sources
I believe that the manuscript lacks statistical analysis, the authors in the discussion even emphasize that MLVA types do not cluster (Line 336) but there is no mathematical confirmation, I believe that a correlation should be established between geographical origin and source of insulation and MLvA type
Conclusion: conclusions are only generalities, missing specific information resulting from the obtained results
References: no italics in Latin names, months just in some references, some like [6] and [7] are not full
Author Response
Dear Reviewers
I have attached a letter addressing all the comments made by the reviewers
Regards
Ayanda

Reviewer 2 Report
The manuscript presents results of MVLV analysis of Listeria species that were identified in Farms and beef meat from retail stores in two South African provinces. It Listeria species can cause severe health problems for humans, therefore their detection and monitoring is of great importance. The authors show that MLVA is a successful method in subtyping of Listeria species providing fast and affordable results.
The manuscript is well written. Introduction provides all necessary background information of the current knowledge on the subject and methods are described with sufficient details. Results are well presented and discussed.
I suggest only to add a full names of abbreviations MLVA and PCR in the text.
In line 136 “g” should be in italic an there should be space between “x” and “g”
Author Response
Dear Reviewers
I have attached a letter addressing all the reviewer's comments
Kind regards
Ayanda Manqele
